**Review**

ammonium; homeostasis; nitrate; nitrogen; transporter.

**Corresponding author:**
Anthony J. Miller;
Email: tony.miller@jic.ac.uk

**Associate Editor:**
Prof. Ingo Dreyer

# Homeostasis of ammonium and nitrate in plants

Yi Chen and Anthony J. Miller

Biochemistry and Metabolism, John Innes Centre, Norwich, UK

## Abstract

Nitrogen (N) is a major plant nutrient, and its supply is very often limiting growth. The main forms of inorganic N in soil supplying plants are ammonium and nitrate ions. Although the soil availability of N can vary greatly, the cytoplasmic nutrient ion activities in a typical plant cell are maintained at set points that are independent of changes in supply. By contrast, the storage of N as protein and vacuolar nitrate depends on the external supply. Measurements of cellular homeostasis of ammonium and nitrate are limited by methodology. The upper limits for cytoplasmic set points are likely to depend on toxicity, and for ammonium this is well known but less clear for nitrate. An intracellular set point for N must be maintained by membrane transport systems and assimilation processes. Crop N use efficiency has uptake and assimilation components, and understanding homeostasis is fundamentally important for improving this important trait.

## 1. Introduction

As plants are unable to move to relocate themselves, they need to be able to adapt and adjust to their constantly changing environment. The plasticity of plants is the key to their success, and this requires an ability for plant cells to maintain an internal stability or 'homeostasis' even when the external conditions vary widely. Cellular homeostasis requires the preservation of a physiological variable at a set point that is an optimal value for physiology, despite external and internal disturbances. The set point concept comes from control theory, requiring a sensing system that detects deviations from the set point. A regulatory network usually acts via negative feedback to restore the variable towards the set point. In this review, we explore the core idea of homeostasis of plant nitrogen (N), more specifically the inorganic N ions ammonium ($NH_4^+$) and nitrate ($NO_3^-$) within cells. We identify some of the control theory components in plants for $NO_3^-$ and $NH_4^+$ that might define N homeostasis. Transport and assimilation are the main drivers for plant N homeostasis, and the balance between these two processes drives growth and yield. These topics have been extensively reviewed previously, and for brevity, recent overviews are cited.

## 2. Whole plant N homeostasis

N is an essential macronutrient for plant growth, and the optimisation of agricultural production depends on the addition of N fertiliser. Chemical fertilisers are often applied with N available as $NH_4^+$ or $NO_3^-$, and other forms of organic N, such as urea, are rapidly converted by soil microbes into these inorganic forms. Plants can take up both inorganic N forms: $NO_3^-$ and $NH_4^+$. Legume plants can capture gaseous $N_2$ through symbioses with N-fixing microorganisms, forming the neutral molecule ammonia ($NH_3$), which can be transferred to the host. Gaseous $NH_3$ can be captured by plants and can diffuse through cell membranes, but this environmental source of N is not usually important for growth (Xu et al., 2012). Organic sources of N in the soil, including amino acids, urea and ureides, can be taken up and used by plants. These forms of N are particularly important for plants growing in impoverished soils. N-starved plants have systemic long-distance signalling mechanisms mediated by phloem-mobile microRNAs (miRNAs) (Liang et al., 2012) and peptides (C-terminally encoded peptide or CEP) in the phloem (Ohkubo et al., 2017). Circulating peptide signals are perceived by specific receptors and can modify strigolactone hormone levels to alter root (branching) and shoot (tillering) architecture. In contrast, N-replete status is mediated by circulating cytokinin hormone levels (Poitout et al.,

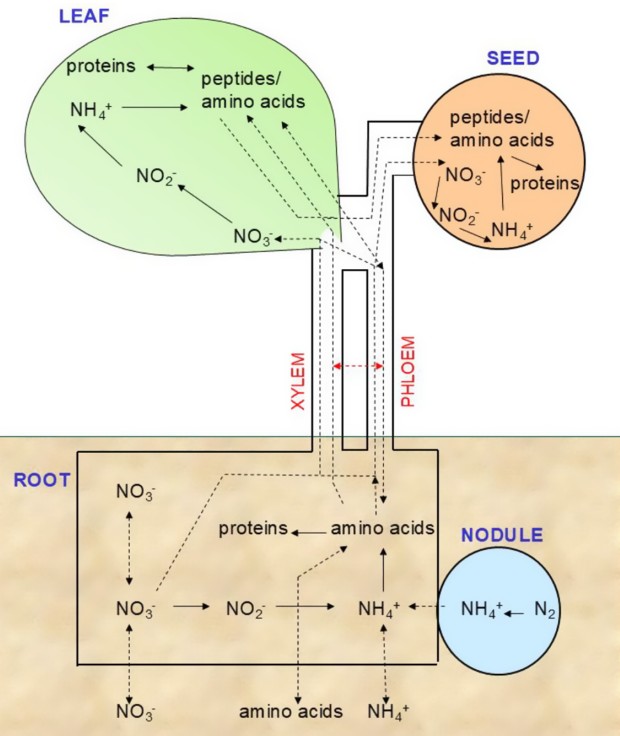

**Figure 1.** Whole plant N homeostasis pools.

2018). Insufficient N levels can lead to restricted plant growth and a shift from the vegetative to the reproductive developmental phase. Interestingly, N deficiency encourages extensive root development, while sufficiency promotes resource allocation for enhanced vegetative growth (Xu et al., 2012). The process of N allocation is heavily influenced by transporter activity, not only through uptake from the soil but also for the movement of N around the plant. During plant development the distribution of N changes as vegetative growth builds the canopy that later switches to florescence and seed filling (see Figure 1). Throughout these developmental changes the tissue pools of inorganic N ions could alter but only $NO_3^-$ changes directly indicate the overall N status of the plant. For this reason, the crop N status testing applied by farmers uses tissue sap $NO_3^-$, not $NH_4^+$ as an indicator ion species when deciding on fertiliser application rates. This highlights the crucial role of $NO_3^-$ as an indicator ion species and therefore identifies it as a signal in coordinating N utilisation and plant growth. Undeniably, $NO_3^-$ signalling triggers the activation of genes associated with various metabolic pathways and developmental processes (reviewed in Zhao et al., 2018).

Whole plant N homeostasis set points are controlled by variables such as metabolism, growth and signalling to stay balanced. Assimilation, uptake from the soil and long-distance transport within the plant in the phloem and xylem are key processes. The pathways and key enzymes for the assimilation of $NO_3^-$ and $NH_4^+$ are well characterised (Xu et al., 2012). Growth and yield are optimised when N assimilation into amino acids and proteins is most active, and this occurs during periods of maximised photosynthetic carbon assimilation. Nitrate uptake by roots and movement of nitrate to the shoot remains high during the light period and decreases slightly at night. The $NO_3^-$ that is taken up during the nightis chiefly used to replenish the leaf $NO_3^-$ pool (Matt et al., 2001). Once transported to the leaf, $NO_3^-$ is primarily stored within the vacuole, accounting for 58–99% of the total $NO_3^-$ pool (Granstedt & Huffaker, 1982).

Therefore, throughout the day and night, the pool of $NO_3^-$ primarily localised in leaf vacuoles, is continually changing and responding to both the soil supply and assimilation activity. Throughout these fluctuations in $NO_3^-$, tissue $NH_4^+$ concentrations usually remain low, indicating a tight coupling between $NO_3^-$ and $NH_4^+$ assimilation, primarily GS activity. The set points for N homeostasis are fundamentally linked to those of other elements; these are often expressed as ratios, such as C:N (reviewed by Fañanás-Pueyo et al., 2025). The C:N ratio changes in tissues during the switch from vegetative to floral ontogeny, from 15-20:1 to 80-100:1, and two conserved kinases (TOR and SnRK1) play key parts in signalling in all eukaryotes. Other elemental ratios have been studied in much less detail, for example, N:P, although these are linked to the type of plant, higher in graminoids than forbs, and linked to stress tolerance (Güsewell, 2004). Long-distance transport is also closely coupled with the need for electrical charge balance and for example, potassium is well known to counter $NO_3^-$ movement in the xylem and phloem (Drechsler et al., 2015).

## 3. Cellular homeostasis of N

The set points for cytosolic homeostasis of $NO_3^-$ and $NH_4^+$ seem to be similar, with both in the low mM range for the cytosol and an order of magnitude greater for the vacuole (see Figure 2). Although the published values can vary hugely over several orders of magnitude for $NH_4^+$ from 0.02 to 0.2 M, depending on the supply and method used (Miller et al., 2001). There can be fundamental metabolic consequences of $NO_3^-$ and $NH_4^+$ use by plant cells that could directly act on intracellular homeostasis through regulatory mechanisms for pH and redox regulation. For example, excessive $NO_3^-$ transport can acidify the cytosol and activate proton pumps

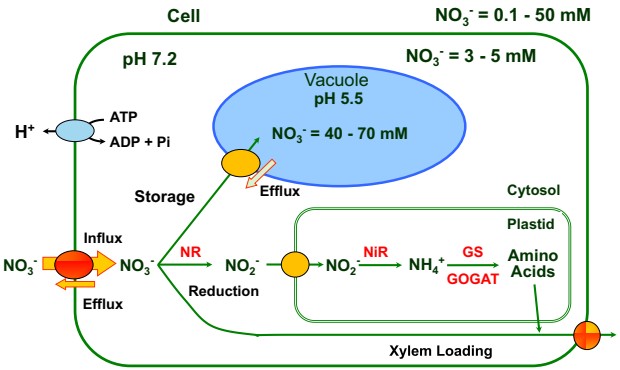

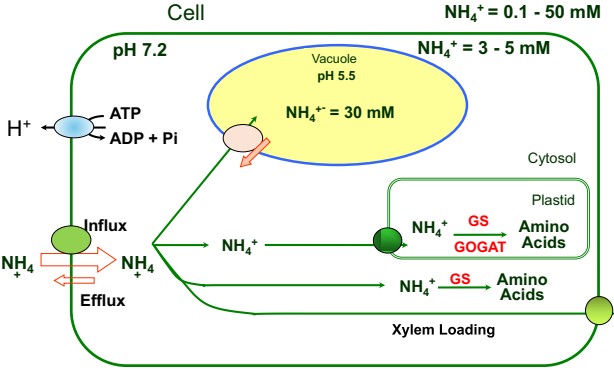

**Figure 2.** Diagrammatic overview of plant cell ammonium and nitrate homeostasis showing metabolism and transport. Upper, nitrate; Lower, ammonium.

to restore pH homeostasis (Feng et al., 2020). Nitrate assimilation consumes reducing power in the cell through NADH- or NADPH-mediated transfer of reductant. In Arabidopsis leaf cells, both photosynthesis and nitrate reductase activity were shown to alter the cytosolic $NO_3^-$ set point (Cookson et al., 2005). These primary assimilatory processes may provide another level of control that is important when there is an excessive supply of either of these forms of N supply.

When plants are N replete, the few measurements which have been made suggest that cytosolic $NH_4^+$ and $NO_3^-$ activities are maintained at mM activities that are independent of the external supply (e.g. $NO_3^-$: Miller & Smith, 2008; $NH_4^+$: Wells & Miller, 2000; Zhou et al., 2015). These homeostatic set point values are higher than the affinities of the enzymes that assimilate the ions; GS/GOGAT for $NH_4^+$ and $NO_3^-$ reductase for $NO_3^-$ (see Figure 2). Suggesting that assimilatory enzyme activities are not directly determining the cytosolic set points. A range of set point values has been reported, depending largely on the type of method used, and as discussed above, more measurements and new techniques are needed.

### 3.1. The evidence for set points of cellular $NH_4^+$ and $NO_3^-$

A previous review (Miller & Smith, 2008) described the evidence that cytosolic $NO_3^-$ homeostasis is regulated and has a potential role in nutrient sensing and signalling. It was suggested that root tips may be particularly important in this sensing role. Nitrate is essential for plant growth and development, even in organs like seeds where its nutritional role is unclear (Chopin et al., 2007). It can act as a signal for germination and shoot–root growth balance. Different cell types and tissues can show varying $NO_3^-$ concentrations, with vacuolar $NO_3^-$ pools acting as reservoirs to maintain cytosolic $NO_3^-$ levels. Twenty years ago, the evidence for these ideas was largely based on measurements using intracellular $NO_3^-$-selective electrodes. Nitrate-selective microelectrodes measurements have shown environmental conditions when cytosolic nitrate changed, for example, during light/dark changes in leaf cells, $NO_3^-$-starvation and when $NO_3^-$ and ammonium or glutamine were supplied together as a mixed supply of N (Miller & Smith, 2008).

The stability of cytosolic $NO_3^-$ is controversial, and even now, the methods to measure it are limited – new techniques are needed, and despite progress in using genetically encoded sensors for other ions (Sadoine et al., 2023), obtaining quantitative $NO_3^-$ measurements remains a tricky nut to crack (*Clophensor* Demes et al., 2020; *NitraMeter3.0* Chen et al., 2022; *NitrOFF* Cook et al., 2025). These genetically encoded sensors are often pH sensitive, but in the cytoplasm the tight regulation of pH should minimise this problem. The high affinity of the bacterial-derived sensors, for example, the nitrate recognition domain NreA from *Staphylococcus carnosus* ($K_d$=9 μM) of *NitrOFF*, should be saturated at the mM concentrations suggested from microelectrode measurements. For a comparison of the properties of genetically encoded sensors for $NO_3^-$, see Table S1 in Cook et al. (2025).

The toxicity of $NH_4^+$ accumulation in biology is widely accepted and therefore the need for plants to regulate their cellular concentration is less controversial (Bittsánszky et al., 2015).

### 3.2. Nitrate and ammonium transporters

Four gene families have been identified as $NO_3^-$ transporters; these are proton-coupled symporters NRT1s (NPFs) and NRT2s (Wang et al., 2012), proton/anion antiporters CLCs and the NAXT/SLAC/SLAH anion channels (Kollist et al., 2011). Many NRT1 (NPF) and NRT2 $NO_3^-$ transporters have been characterised in detail, they are proton-coupled cotransporters that can operate efficiently over a large range of $NO_3^-$ concentrations (reviewed in Wang et al., 2012; Xu et al., 2012). Two families of $NH_4^+$ transporters (AMT1 and 2) are well described and, like the NRTs, also have wide-ranging affinities for $NH_4^+$ (Xu et al., 2012). The entry of $NH_4^+$ into cells can occur via non-selective cation channels, driven by the large negative membrane potential across the plasma membrane (Zhou et al., 2015). Both types of transporters, $NO_3^-$ and $NH_4^+$, have proton-coupled mechanisms to drive uptake from low external concentrations of the ions. In Figure 2, we have summarised the main cellular pathways for both ions, $NO_3^-$ and $NH_4^+$. At the cell plasma membrane, we show efflux mechanisms for both ions, but the precise genetic identity is not clear. Stretch-activated anion channels for $NO_3^-$ and outwardly directed cation channels for $NH_4^+$ have been functionally characterised (Hedrich, 2012). At the plasma membrane, nitrate efflux is mediated by AtNRT1.5,1.8,1.9 (Wang et al., 2012), SLAC/SLAH anion channels, and these are important for xylem loading and stomata guard cell function (Hedrich & Geiger, 2017). The aluminium-activated malate transporters are generally considered organic anion channels, but some family members are permeable to $NO_3^-$ (e.g. rice OsALMT7, Heng et al., 2018).

In addition to regulation of transcript levels, both types of transporters, $NO_3^-$ and $NH_4^+$, have post-translational mechanisms (e.g. phosphorylation) to regulate their activity (see Section 4) and this fact supports the view that transport is a key factor in plant N homeostasis (e.g. $NO_3^-$: Yue et al., 2025; $NH_4^+$: Wang et al., 2021).

### 3.3. Vacuolar transporters

There are three protein families involved in $NO_3^-$ transport through the vacuolar membrane, CLCs (De Angeli et al., 2006; Yang et al., 2023), NRT2s (e.g. Chopin et al., 2007) and NPFs (e.g. He et al., 2017). Plant tissue $NO_3^-$ concentrations directly follow the storage concentrations in the vacuole, which in turn generally reflect the changing levels in the soil. Remobilisation of vacuolar-stored $NO_3^-$ is important for NUE (Chen et al., 2020) and this is an important target for improving NUE that may also have consequences for water use efficiency as vacuolar $NO_3^-$ is an important osmoticum (Hodin et al., 2023). In Arabidopsis, vacuolar AtCLCa activity is regulated by phospholipids and nucleotides like ATP (Yang et al., 2023).

Information for $NH_4^+$ transport at the vacuole is complicated by the fact that $NH_3$ is trapped in this acidic compartment. Ammonia, due to the pH gradient between the cytoplasm and vacuole, diffuses across the tonoplast into the acidic vacuole, where it is trapped as the charged $NH_4^+$ ion in the acid compartment. In the vacuolar membrane, the tonoplast intrinsic proteins (TIPs) that can mediate water movement have also been shown to facilitate $NH_3$ transport into the vacuole (Loqué et al., 2005). Both $NH_4^+$ and $NH_3$ may be exported from the vacuole with the possibility for post-translational regulation of the direction of transport. This topic is important and needs more research effort.

## 4. Regulatory network: restoration of the set point

We can accept and build on the tenet that all homeostatic control mechanisms have a minimum of three interdependent components for the variable being regulated: a receptor, a control centre and an effector. This textbook homeostatic model has been formulated

for animal cells (Billman, 2020). Applying these fundamental questions to the plant cell model, we can look for the three components.

### 4.1. Cell N sensing

Are there receptors for $NO_3^-$ or $NH_4^+$ in plant cells? The transporter NRT1.1 has been described as a $NO_3^-$ transceptor linked to calcium signals via kinase activity (Wang et al., 2012). A receptor should have some capacity to specifically bind either ion with an affinity that perhaps matches the range of possible homeostatic values of the cytosolic activities of these ions. Nitrate-binding proteins that are regulators of transcription have been identified in the nucleus, for example, NLP7 (Cheng et al., 2023; Liu et al., 2022). The nucleus may seem an odd place to perceive external environmental changes in N supply, but if there is homeostasis in the cytosol and this depends on the plant's N status, then this makes sense. The nitrate-binding domain of NLP7 has been identified and a genetically encoded fluorescent split biosensor, mCitrine-NLP7, enabled visualisation of single-cell nitrate dynamics *in planta* (Liu et al., 2022). As amino acids are the primary product of N assimilation, they have been proposed as potential negative feedback signals for N status. To this end, plant glutamate receptor-like genes have been identified as $Ca^{2+}$ channels that bind amino acids, but their functions appear more widespread, including long-distance electrical signalling and redox sensing (Simon et al., 2023).

There are many examples of post-translational modifications of proteins involved in the transport and assimilation of $NH_4^+$ and $NO_3^-$, this tight regulation provides indirect evidence for homeostasis and set points in the N physiology.

### 4.2. Regulation of assimilation enzymes

The activity of the primary N assimilatory enzymes is carefully regulated by phosphorylation; for example, NR is regulated by phosphorylation and subsequent 14-3-3 protein binding (Lambeck et al., 2012) and GS (Finnemann & Schjoerring, 2000). These are enzymes that catalyse key steps in N assimilation and homeostasis, as the former generates toxic nitrite ($NO_2^-$) and the latter is the confluence of C and N assimilation. NR activity can generate nitric oxide (NO) when $NO_2^-$ concentrations increase, and this reactive oxygen species can result in S-nitrosylation, reacting with cysteine thiols in proteins, an indicator of stress, redox status and an imbalance in N homeostasis. This reaction can also regulate protein SUMOylation linking N status to plant immunity (Borrowman et al., 2023). This mechanism is a reversible post-translational modification where SUMO (small ubiquitin-like modifier) proteins are covalently attached to lysine residues on target proteins.

### 4.3. Regulation of $NH_4^+$ and $NO_3^-$ transporters

Phosphorylation of transporters is a common regulatory mechanism and there are many examples (Hao et al., 2023). The calcium-dependent protein kinase and CIPK families of plant protein kinases function in calcium signalling pathways and are important in the regulation of N cellular homeostasis. Both $NH_4^+$ (AMT1.1/2) and $NO_3^-$ (NRT1.1) transporter activity in the model plant Arabidopsis are regulated by the activity of CIPK23 (reviewed by Ródenas & Vert, 2020). At high $NH_4^+$ concentrations, the AMT transport activity is inactivated by phosphorylation. In contrast, at low $NO_3^-$ concentrations, AtNRT1.1 transport activity is activated by CIPK23 phosphorylation. There is allosteric regulation of plant AMTs, involving a phosphorylation switch that functions in a feedback loop to restrict $NH_4^+$ uptake (Lanquar et al., 2009).

The activity of NRT2 transporters is also regulated by phosphorylation, including their interaction with the Nar2 partner proteins that are required for function (Jacquot et al., 2020; Li et al., 2024). Taken together there are multiple levels of control for the entry of inorganic N into non-leguminous plants. In legumes, the situation is further complicated by the activity of the symbiotic bacteria living in nodules and generating $NH_3$ from gaseous atmospheric $N_2$.

### 4.4. Regulation of vacuolar transporters

All three protein families of vacuolar $NO_3^-$ transporters can be post-translationally regulated. In Arabidopsis, vacuolar AtCLCa $NO_3^-$ selectivity is determined by a proline residue (Wege et al., 2010) and activity is regulated by phospholipids and nucleotides like ATP (De Angeli et al., 2009; Yang et al., 2023). The vacuolar CLCs also have an important role in cytosolic pH regulation (Demes et al., 2020), which is key for pH balance during the switch between $NH_4^+$ and $NO_3^-$ external supply. This provides an interesting direct link between N supply, homeostasis and pH regulation that is worthy of future investigation. The activity of the aquaporin TIPs that facilitate $NH_3$ transport (Loqué et al., 2005) has also been shown to be regulated by phosphorylation of the protein (Maurel et al., 1995). The direction of tonoplast transport of $NH_4^+$ and $NH_3$ may be post-translationally regulated and vacuolar accumulation of $NH_4^+$ is regulated by CAP1 (Bai et al., 2014).

### 4.5. Transcription factors

Plant transcription factors (TFs) are important molecular regulatory systems for N homeostasis, typically comprising 5% of the genome (Blanc-Mathieu et al., 2024), they coordinate N transport and assimilation (e.g. $NO_3^-$, Sámano et al., 2024). There are several well-known examples, for example, HY5 is important for C:N partitioning (Chen et al., 2016). The NIN-like proteins (NLPs) are a group of TFs belonging to the RWP-RK gene family, they act as major nitrate sensors and are implicated in the primary $NO_3^-$ response within the nucleus of both non-leguminous and leguminous plants through their RWP-RK domains. The NLPs can act as intracellular nitrate sensors, as they can bind $NO_3^-$ and thereby alter the expression of other transcripts linked to the N status of the cell. In the model plant, Arabidopsis, there are nine *NLP* genes expressed in *Arabidopsis* shoots, and the role of some family members in binding $NO_3^-$ to alter transcription is well established (Sámano et al., 2024). In a legume, $NO_3^-$ induces SUMOylation of the NLP1 TF (Liu et al., 2026). This mechanism may yet prove to be more widely found among plants, providing a link between N homeostasis and TFs.

As TFs are more general regulators of N acquisition and homeostasis, they have been targets for efforts to improve crop fertiliser use (Maurya et al., 2020). There are now many published examples of altered expression of N-responsive TFs that have been used in crops to have significant effects on NUE (see Table 1).

## 5. Future perspectives and N use efficiency

Improving N use efficiency (NUE) is the target of major research efforts. Plant N homeostasis and NUE are interdependent and closely linked. Altering N set points is a key component of attempts to improve NUE and here an opportunity can be provided by natural ecosystems as research using rapidly growing species adapted to nutrient-poor environments signpost the route to better crops. As described above, TFs are targets for altering C:N ratios and NUE as they offer a high tier of control (Figure 3) and there are

**Table 1.** Some N-related TFs used in crop plants to improve NUE (redrawn from Maurya et al., 2020)

| Transcription factor family | Gene | Target species | Physiological and phenotypic responses | Reference |
|---|---|---|---|---|
| MADS-box | OsMADS57 | Rice | Increased expression of $OsNRT2.1/2.2/2.4$ and $OsNRT2.3a$Increased root to shoot N remobilisation, Shoot-root $^{15}$N content ratios were increased up to 76% at low $NO_3^-$ supply | Huang et al., 2019 |
| | OsMADS25 | Rice | Increased biomass, PR length, lateral root no. and length in the presence of $NO_3^-$ | Yu et al., 2015 |
| NLP | OsNLP1 | Rice | Increased growth, grain yield and NUE up to 20.5% under low $NO_3^-$ supply | Alfatih et al., 2020 |
| | OsNLP4 | Rice | Increased biomass, grain yield (30%) and NUE (47% moderate N supply) | Wu et al., 2021 |
| | ZmNLP5 | Maize | Regulates nitrite reductase expression. Knock-out accumulated less seed N. | Ge et al., 2020 |
| b-ZIP | TabZIP60 | wheat | Increased NADH-dependent glutamate synthase (NADH-GOGAT) activity Increased lateral root branching, N uptake and spike number and improved grain yield >25% in field trial | Yang et al., 2019 |
| NAC | OsNAP | Rice | Increased N content in grain and flag leaf, higher grain weight and yield up to 10.3%. Higher seed-setting rate Delayed senescence | Liang et al., 2014 |
| | NAM-B1 | Arabidopsis wheat | Increase in N remobilisation from leaves to grain. Increased grain protein and mineral content ~30% | Uauy et al., 2006 |
| | Ta NAC2−5A | Wheat | Increased expression of TaNRT2.3B and GS2 Increased biomass and grain yield (up to 10% at low $NO_3^-$ level). Increased root $NO_3^-$ influx rate | Li et al., 2020 |
| MYB | SiMYB3 | Millet, Arabidopsis and rice | Increased expression of NRT1.1, NIA2, ANR1, NLP7, LBD37, LBD38, LBD39, TAR2 and IPT3 Increased grain weight, total and seed N and root growth under low N, P and K. | Ge et al., 2019 |
| | OsMYB305 | Rice | Increase in expression of OsNRT2.1, OsNRT2.2, OsNAR2.1, OsNiR2, NADH-GOGAT, Pyr-K and G6PDH under low N conditions Increased tillering, shoot dry weight and total N concentration. Increased 15NO3$^-$ influx | Wang et al., 2020 |
| Dof | RDD1 | Rice | Increased $GS1;1$ expression Increased chlorophyll content $NO_3^-$ and $NH_4$- uptake under limited N supply. Higher grain harvest index and yield. | Iwamoto & Tagiri, 2016 |
| | ZmDof1 | Rice | Increased PEPC activity and asparagine content. significant increases in the rate of photosynthesis, increases in the amounts of both N and C per seedling, increased root N and biomass under limited N supply | Kurai et al., 2011 |
| | ZmDof1 | Wheat and Sorghum | Increased PEPC activity and amounts of N and C per seedling. Higher biomass production under up to 59% limited N supply | Peña et al., 2017 |
| NF-Y | TaNFYA-B | Wheat | Higher expression of transporters NRT1.1, NRT2.1 and some phosphate transporters. Increased Lateral root growth and root $NO_3^-$ influx. Increased tissue $NO_3^-$, grain yield spike number under differing N supply levels | Qu et al., 2015 |

many candidates (Table 1). It remains to be tested to what extent the effects of each TF may be additive, and this may provide a breeding opportunity. Both N assimilation and uptake are important components of NUE. The assimilation of N is limited by the photosynthetic production of sugars for amino acid synthesis by GS/GOGAT activity. The energy from carbon metabolism and supply of reductant from photosynthesis power the uptake and assimilation of N. Therefore, plant N homeostasis and NUE are directly linked to carbon assimilation and future efforts to improve crops must recognise this fact and integrate both traits into breeding programmes. The supply of N to cells, maintaining an optimal concentration for assimilation, is a fundamental driver for homeostasis both at the cellular and whole plant levels. If N homeostasis breaks down this is likely to be very detrimental for growth, but when it is sub-optimal then crop productivity is below ideal. Breeding future crops to improve the plasticity of N homeostasis is likely to be a promising strategy, particularly with the threat of climate change extreme weather events. Therefore, a better understanding of plant N homeostasis and what can limit it in cells and the whole plant is fundamentally important for improving NUE. A holistic view that includes N and C cycling is important for this progress.

In the future, a better understanding of factors extending beyond the plant, into the rhizosphere and the soil, that influence N homeostasis is important. For example, soil C supply is important

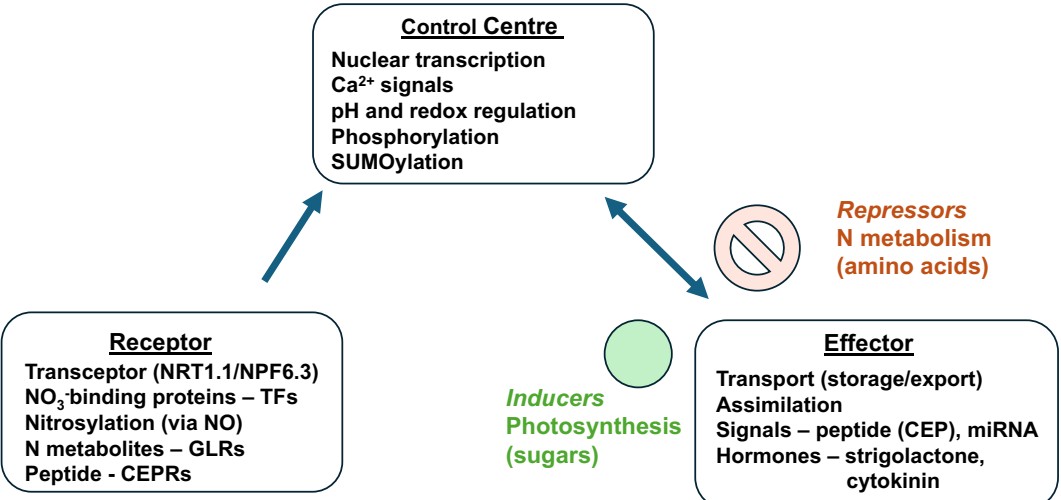

**Figure 3.** Identifying the model components of plant N homeostasis.

for nutrient cycling and the slow release of N for crops. Maintaining a slow, steady N supply that matches the changing needs of the crops and balancing homeostasis can directly depend on root activity. Root exudates can modify soil microbial activities to influence the form of N supply and optimise growth. Optimising the balance in the N supply, $NO_3^-$ and $NH_4^+$ (Feng et al., 2020) for the crop during plant development is likely to be an important aspect of improving crop NUE. To achieve this soil delivery target, better methods for monitoring soil nutrient delivery are needed. There are also exciting opportunities for phage engineering of the rhizosphere microbiome (Quirós et al., 2023) to better optimise microbially mediated N cycling to balance the soil supply of $NO_3^-$ and $NH_4^+$ to crops. A holistic view of N homeostasis should extend beyond the whole plant and cells into the soil for improving NUE.

**Open peer review.** To view the open peer review materials for this article, please visit http://doi.org/10.1017/qpb.2026.10042.

**Abbreviations and Glossary.**

| | |
|---|---|
| ALMT | Aluminium-activated Malate Transporter |
| CAP1 | Adenylate Cyclase-Associated Protein 1 (actin-binding protein) |
| CBL | Calcineurin B-like protein |
| CEP | C-terminally encoded peptide |
| CEPR | CEP Receptor |
| CIPK | CBL-interacting protein kinase |
| CPK | Calcium-dependent protein kinase |
| GOGAT | glutamate synthase |
| GS | glutamine synthetase |
| NAD | Nicotinamide Adenine Dinucleotide |
| NADPH | nicotinamide adenine dinucleotide phosphate |
| NIN | Nodule inception TF |
| NiR | nitrite reductase. |
| NLP | NIN-like proteins |
| NR | nitrate reductase |
| RWP-RK | a family of TFs |
| SLAC | Slow-type anion channel |
| SLAH | SLAC-1 homolog anion channel |

| | |
|---|---|
| SnRK1 | SNF1-related protein kinase 1, a key cellular energy-sensing protein kinase. |
| SUMO | Small Ubiquitin-like Modifier |
| TF | transcription factor – a highly conserved serine/threonine protein kinase |
| TOR | Target of rapamycin |

**Data availability statement.** No new data or coding was used in preparing this manuscript.

**Acknowledgements**

Y.C. and T.M. are funded by the BBSRC Institute Strategic Programmes: Plant Health (grant no. BB/P012574/1) and Advancing Plant Health (grant no. BB/X010996/1); Harnessing Biosynthesis for Sustainable Food and Health (grant no. BB/X01097X/1).

**Author contributions.** Y.C. and T.M. planned and wrote the review.

**Funding statement. WishRoots:** The BBSRC Wish-Roots 21EJP Soil: Tuning the wheat root microbiome to improve soil health and optimise rhizosphere nitrogen cycling and availability (grant no. BB/X003000/1). **APH:** The BBSRC Institute Strategic Programmes: Plant Health (grant no. BB/P012574/1) and Advancing Plant Health (grant no. BB/X010996/1). **HBio:** The BBSRC Institute Strategic Programme: *Harnessing Biosynthesis For Sustainable Food and Health* (grant no. BB/X01097X/1).

**Competing interest.** The authors declare no conflict of interest.

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
