## [Reviewer Report]

This manuscript by the authors summarizes recent findings on ammonium and nitrate homeostasis in plants. The review is well-organized and easy to read. It is structured to help newcomers to the field understand current trends. I have commented on minor points, particularly regarding the presentation of figures. I hope my comments will further enhance the quality of this work.

Line 97, 200 Instead of Figure 1AB, simply Figure 1 should be fine.

Line 139-140 citation required.

Line 201, from 0.2 to 0.2 M does not make sense. 0.2 mM to 0.2 M? or 0.02 to 0.2 M?

Figure 1B

GS is known to localize to the cytoplasm and chloroplasts, but does GOGAT localize to the cytoplasm? Aren’t both Fd-type and NADH-type forms chloroplast-localized?

Is there no pathway for ammonium trapped in the vacuole to be expelled? In reality, since TIP is localized in the vacuole, I think a certain proportion of NH3 present in the acidic vacuole interior is transported out of the vacuole. What is the author’s view on this?

In Figure 1B, ammonium is loaded into the xylem, but in Figure 2, ammonium is converted to amino acids before being transported by the xylem. The figures seem to contradict each other regarding the possibility of ammonium entering the xylem. How should this be interpreted? Also, while only nitrate is shown being transported from roots to shoots via phloem, is there no possibility that amino acids could be transported to the shoot by phloem?

In Figure 2, nitrate ions appear twice in the leaf. What is the difference between the two instances?

Are nitrate and ammonium not present within the seed?

---

## [Reviewer Report]

This review manuscript aims to provide an overview of the mechanisms underlying nitrate and ammonium homeostasis in plants. While this topic is highly relevant in plant physiology, the manuscript in its current form exhibits limitations in terms of depth, novelty, and narrative structure, which considerably reduce its potential contribution to the field.

1) The title sets the focus on nitrate and ammonium homeostasis and the abstract defines set points as important hallmarks to evaluate homeostasis. However, the core and relevance of the set point concept are not explained. Instead, the manuscript reiterates general descriptions of transport, assimilation, and regulation that have been extensively covered in previous reviews.

2) I really miss a synthesis that clearly defines the concept of homeostasis alos between ammonium and nitrate or N forms and pH, highlights recent advances, and identifies drivers or determinants of homeostatic control.

3) The discussion about homeostatic control mechanisms (line 183-196) is incomplete and weird. It remains unclear to what extent the authors propose that the known components (NRT1.1, NLP7 etc.) fulfil the homeostatic model requirements and whether this model is of use to reflect or even explain plant responses to varying N regimes.

4) Chapter 4 on the transcription factors is poorly integrated into the topic and does not contribute explaining homeostatic control mechanisms.

5) The consideration of relevant literature is poor. For instance, I miss regulatory mechanisms of vacuolar ammonium by CAP1 (Bai et al. 2014, Plant Cell). Instead, the text provided at line 133-140 is leading away from the topic. Similar for nitrate.

6) The ‘future perspectives’ section is also very limited and does not propose new directions or provide a robust conceptual framework. A central statement says that ‘plant N homeostasis and NUE are directly linked to carbon assimilation and future efforts to improve crops must recognise this fact and breed for both traits.’ The message here is completely unclear: What trait shall serve as read-out for homeostasis? How will homeostasis improve NUE? Also a NUE-inefficient plant can be in full homeostasis. I consider these conclusions as premature.

7) Finally, I recommend that the authors update the bibliography, improve the quality of English writing, and correct typographical errors, such as in line 47, ‘Nitrogen (N) is an essential micronutrient for plant growth…’, which are not acceptable in a review of this type.

---

## [Reviewer Report]

The present manuscript describes the mechanisms regulating the cellular and whole plant homeostasis of nitrate and ammonium. However, the aim of this review is not clear. Indeed, dose it aims to summarize the functions and regulation of different transporters in ionic homeostasis or does it aims at describing the ionic homeostasis at a higher level of integration. In its current form, the review is unclear and consequently does not bring significant information. Further, it is overall not cite the correct bibliography and it is missing many key papers. The concepts and findings are presented in a confusing and sometimes wrong way. I regret to say that I do not recommend its publication in the current form and that would require massive modifications to reach a publication grade.

Some major points are listed below:

- The abstract is unclear the sentences between line 28 and 34 are not logically connected. It sounds like sentence side to side without connections.

- Line 78-82. Concerning the biosensors the authors forgot many among with Clophensor (Demes et al. 2020) NitrOFF (cook et al. 2025).

- Nitrate transporters paragragraph. There is nothing on NRT1 and NRT2 apart from a citation. line 93 and 124 CLC lacks a reference and there more recent and complete reviews about CLC that Ziffarelli&Pusch 2027.

- Line 98. The SLAC/SLAH channels are nitrate efflux channels at the PM, the group of R. Hedrich and D. Geiger did a lot of work on this. No direct citation for NH4+ channels.

- Line 117. There is a lack of refs about NRT2 phosphorylation such as Jaquot et al. 2020

- Line 130. It is not clear why the ATP regulation of CLCa is mentioned here and the reference should include De Angeli et al. 2009 that was the first paper on it. Why the origin of the NO3 selectivity is not mentioned (De Angeli et al. 2006; Wege et al. 2010;) ? It would maybe be more informative.

- Line 139. No reference about TIP regulation by phosphorylation.

---

## [Editor Report]

Dear Dr. Miller,

Thank you for your contribution to the Research Topic “Quantitative approaches to cellular aspects of plant ion homeostasis”. Your manuscript has been assessed by 3 reviewers. As you can see from their comments, they raise important (and in some cases overlapping) points of criticism, which could still be addressed as part of a comprehensive, substantial revision. Alternatively, we may also consider a re-submission of a fully revised manuscript on that topic.

Best regards, Ingo Dreyer

---

## [Reviewer Report]

This is a very well constructed short review of various aspects of nitrogen homeostasis. Importantly, quantitative consideration is given to the set points for nitrate and ammonium in the cytosol and vacuole, as well as the factors controlling these set points. The review also valuably highlights areas where there are gaps in our knowledge and where there is a need for further research.

There is, however, a need for the authors (or a copy editor) to undertake corrections to the manuscript presentation. In several places (e.g. lines 30, 73, 136, 265) sentences have no active verb, there are typographical errors (e.g. line 281), and commas appear seemingly at random. Once these corrections are made, I believe the MS should be in good shape for publication.

---

## [Reviewer Report]

This is a good overview article on the role of nitrogen (nitrate and ammonium) in plants. It could also be mentioned that channels from the ALMT family are also permeable to nitrate. Furthermore, a careful polishing of the text would be very helpful.

---

## [Editor Report]

Dears Tony and Yi,

you MS has been seen by two independent reviewers. Both consider it suitable for publication after some final minor revisions. In particular, a careful language check would be highly appreciated. Thank you for your valuable contribution to the Research Topic “Quantitative approaches to cellular aspects of plant ion homeostasis”. Best regards, Ingo

---

## [Editor Report]

Dears Tony and Yi,

thank you for the careful revision of the manuscript and thanks again for your valuable contribution to the Research Topic “Quantitative approaches to cellular aspects of plant ion homeostasis”. It is highly appreciated.

Best regards, Ingo